



# Coupling of a sediment diagenesis model (MEDUSA) and an Earth system model (CESM1.2): a contribution toward enhanced marine biogeochemical modelling and long-term climate simulations

Takasumi Kurahashi-Nakamura[1], André Paul[1], Guy Munhoven[2], Ute Merkel[1], and Michael Schulz[1]

[1]MARUM - Center for Marine Environmental Sciences and Faculty of Geosciences, University of Bremen, Bremen, Germany
[2]Laboratoire de Physique Atmosphérique et Planétaire, Université de Liège, Liège, Belgium

**Correspondence:** Takasumi Kurahashi-Nakamura (tkurahashi@marum.de)

**Abstract.** We developed a coupling scheme for the Community Earth System Model version 1.2 (CESM1.2) and the Model of Early Diagenesis in the Upper Sediment of Adjustable complexity (MEDUSA), and explored the effects of the coupling on solid components in the upper sediment and on bottom seawater chemistry by comparing the coupled model's behaviour with that of the uncoupled CESM having a simplified treatment of sediment processes. CESM is a fully-coupled atmosphere-

ocean-sea ice-land model and its ocean component (the Parallel Ocean Program version 2, POP2) includes a biogeochemical component (BEC). MEDUSA was coupled to POP2 in an off-line manner so that each of the models ran separately and sequentially with regular exchanges of necessary boundary condition fields. This development was done with the ambitious aim of a future application for long-term (spanning a full glacial cycle; i.e., $\sim 10^5$ years) climate simulations with a state-of-the-art comprehensive climate model including the carbon cycle, and was motivated by the fact that until now such simulations have

been done only with less-complex climate models. We found that the sediment-model coupling already had non-negligible immediate advantages for ocean biogeochemistry in millennial-time-scale simulations. First, the MEDUSA-coupled CESM outperformed the uncoupled CESM in reproducing an observation-based global distribution of sediment properties, especially for organic carbon and opal. Thus, the coupled model is expected to act as a better "bridge" between climate dynamics and sedimentary data, which will provide another measure of model performance. Second, in our experiments, the MEDUSA-

coupled model and the uncoupled model had a difference of 0.2‰ or larger in terms of $\delta^{13}$C of bottom water over large areas, which implied potential significant model biases for bottom seawater chemical composition due to a different way of sediment treatment. Such a model bias would be a fundamental issue for paleo model–data comparison often relying on data derived from benthic foraminifera.



# 1 Introduction

For Earth system models, the simulation of biogeochemical cycles in the ocean is of fundamental importance. Simulating biogeochemistry is important for the projection of unknown (e.g., future) climate states in a prognostic way, because the bio-geochemical cycles play an active role in the climate system by changing greenhouse-gas concentrations in the atmosphere

particularly through the carbon cycle. Secondly, biogeochemical tracers are an important indicator of water masses, and thus a measure of the model quality in representing ocean structures when comparing model states with observations and recon-structions. The distribution of biogeochemical matter in the ocean is determined by internal processes (e.g., physical volume transport, mixing of seawater, and the biological pump) and processes at the upper and lower boundaries. The latter factors, boundary conditions in terms of numerical modelling, consist of two aspects (e.g., Kump et al., 2000; Ridgwell and Zeebe,

2005): the inflow of matter as riverine input following chemical weathering on land (i.e, the upper boundary condition), and the net outflow of marine matter at the ocean floor into sediments (i.e., the lower boundary condition). This study focused on explicitly simulating the process at the lower boundary, that is, the exchange of biogeochemical matter between the seawater and ocean-floor sediments, which motivated the coupling of a climate model including an ocean biogeochemical component and a process-based sediment model dealing with early diagenesis in sediments. Coupling a sediment model is expected to

lead to a better simulation of the sea-water isotopic composition especially for bottom water by providing more realistic lower boundary conditions to an ocean model. This would be particularly important when the climate model is applied to various climate states because a substantial amount of paleoceanographic data is provided by isotopic measurements. In addition to the effects on seawater chemistry, the sediment model will allow to compare models and sedimentary records directly.

    To simulate the sedimentary diagenesis, different modelling approaches with a variety of complexity have been used for

paleoclimatological or global biogeochemical studies (Soetaert et al., 2000; Hülse et al., 2017), which includes a reflective boundary approach where all the sinking particles that reach the sediments are dissolved in the deepest ocean grid cells (e.g., Yamanaka and Tajika, 1996; Marchal et al., 1998), a vertically-integrated reaction layer approach (e.g., Goddéris and Joachim-ski, 2004; Ridgwell and Hargreaves, 2007), and an approach with a vertically-resolved transient diagenetic model (e.g., Heinze et al., 1999; Munhoven, 2007; Tschumi et al., 2011). Up to the present, no fully-coupled comprehensive climate model has

been coupled with a sediment diagenesis model for longer time-scale applications (e.g., the glacial-interglacial variations). In this study, we aim at advancing earth system modelling by coupling a state-of-the-art comprehensive climate model including an ocean general circulation model with a vertically-resolved sediment model dealing with fully coupled reaction-transport equations. Here we report on technical aspects of the coupling and assess the immediate influence of sedimentary processes on the bottom-water chemistry. The assessment is important because it reveals possible model errors or biases of marine bio-

geochemical simulations that depend on the treatment of sedimentary processes in a climate/ocean model. We also discuss possible future applications of the coupled model for studies dealing with the long-term evolution of climate.



## 2 Methods

### 2.1 Models

For the climate part of the coupled model, we employed the Community Earth System Model (CESM, Hurrell et al., 2013; version 1.2). CESM1.2 is a fully-coupled atmosphere-ocean-sea ice-land model, and the ocean component is the Parallel Ocean Program version 2 (POP2). In this study, POP2 was configured to include the Biogeochemical Elemental Cycling model (BEC; Moore et al., 2004, 2013; Lindsay et al., 2014). The BEC model is a nutrient phytoplankton zooplankton detritus (NPZD)-type marine ecosystem model and contains a parameterized routine for sinking processes of biological particles including particulate organic matter (POM) with the "ballast effect" based on Armstrong et al. (2002). The BEC model also includes a highly simplified empirical treatment of dissolution of particulate matter at the ocean floor. For particulate organic carbon (POC) and opal, the amount of dissolved matter given back to the seawater is empirically determined based on Dunne et al. (2007) for POC and Ragueneau et al. (2000) for opal, and the residual matter corresponding to burial is simply lost from the model domain. As to calcite, all particles that reach the ocean floor above a prescribed depth level (3300 m) are lost from the model domain, and all particles dissolve below that level. This original version of CESM with the simplified model for particle dissolution at the ocean floor was used for comparison with the CESM coupled to our sediment module.

The ocean model was also extended with the carbon-isotope component developed by Jahn et al. (2015), so that we were able to simulate explicitly the carbon-isotope composition of seawater, which is an important biogeochemical tracer from a paleoceanographic viewpoint. In this study, a low-resolution setup of CESM following Shields et al. (2012) was used. The ocean component had a nominal 3° irregular horizontal grid with 60 layers in the vertical, while the atmosphere component, the Community Atmosphere Model version 4 (CAM4), had a T31 spectral dynamical core (horizontal resolution of 3.75°) with 26 layers in the vertical. The comparatively fine vertical resolution of POP2 (e.g., 200 to 250-m resolution for the depths deeper than 2000 m) allowed a better representation of bottom water properties, which was particularly important in this study's context. Although only the low-resolution case is shown in this study, a similar procedure will be applicable also to a higher resolution configuration for future applications.

For the sediment model part, we adopted the Model of Early Diagenesis in the Upper Sediment of Adjustable complexity (MEDUSA; Munhoven, 2007). Note that this model is different from the marine ecosystem model with the same acronym (Yool et al., 2013). MEDUSA is a transient one-dimensional advection-diffusion-reaction model that, in its original set-up, describes the early diagenetic processes that affect carbonates and organic matter (OM) in the surface sediment of 10-cm thickness (see also Fig. 2 in Munhoven, 2007). In the 10-cm-thick surface sediment, solids are transported by bioturbation and advection, and solutes by molecular diffusion. Solids that get advected across the lower boundary of this 10-cm-thick sediment get preserved (buried) on a stack of 1-cm-thick layers that can be interpreted as a sediment core. That layered stack is bi-directonally coupled to the overlying reactive mixed layer. Previously buried material can be chemically eroded and returned to the reactive mixed layer (the most recently buried material first) in case boundary conditions evolve in such a way that the thickness of the mixed layer would reduce to less than 10 cm. In the coupled model, one MEDUSA column was coupled to the deepest grid cell of each POP2 water column and there was no lateral exchange of information among the MEDUSA columns.





In this study, MEDUSA was configured such that it treated explicitly eight solid components and nine solute components (Fig. 1), which is a substantial enhancement compared to the original setup in Munhoven (2007). For the calcium-carbonate species, only calcite was taken into consideration in line with the BEC model, although MEDUSA is able to deal with aragonite as well. The time evolution of those chemical species was governed by five processes (Fig. 1): the oxic and suboxic degra-

dation of POM, calcite dissolution, opal dissolution, and the radioactive decay of $^{14}$C. Subject to boundary conditions (i.e., downward solid fluxes, solute concentrations, and physical properties) at the top of the sediment stack, the model forecasts the vertical profiles of solid and solute components in 21 vertical layers. The solute concentrations in deepest grid cells of POP2 were explicitly provided to MEDUSA except for calcium-ion concentration, which was empirically derived from the salinity. MEDUSA then returns solute fluxes at the sediment–water interface back to the water column.

## 2.2   Coupling procedures

The communication between the two models was done in a so-called "off-line" manner; that is to say, we kept the executables of both models separate and exchanged necessary information for their boundary conditions through file exchange. We adopted the off-line coupling considering the much longer characteristic timescale of the sediment model (e.g., a model time step in Munhoven (2007) is 100 years) compared to that of the climate model. The off-line method allowed manageability of model

development and maintenance while physically credible at the same time . However, although we could keep a substantial portion of the original structure of each model especially for the technical procedures (e.g., interfaces to drive the models and input/output routines), it was still required for us to make major alterations to both models and to develop new routines to realize the coupling, as follows:

**Modifying POP2**

– Introduction of new variables for the matter exchanged with MEDUSA as shown in Fig. 1

   – Adjusting writing/reading routines for boundary conditions describing the additional variables

   – Modifying source/sink terms in the tracer prognostic equations for the bottom grid cells of the ocean model

   – Changing the formulation of the boundary conditions at the ocean floor for the particulate matter

**Modifying MEDUSA**

– Creating writing/reading routines for boundary conditions

   – Unit conversion for variables to be exchanged with POP2

**Interfaces between the two models**

   – Format adjustment for the input/output files to utilize the existing schemes of both models as much as possible

   – Automation of procedures: routines for each step of one-time coupling, and a wrapper-level routine to repeat them





For the coupling, CESM and MEDUSA were run sequentially as in the coupling between the atmosphere and ocean components of CESM (Craig et al., 2012); that is to say, each model was driven based on the state calculated by the other in the previous integration period. Otherwise, we would have needed an iterative way to obtain a convergence of fluxes between the two models that satisfied both of them for the same time period. That would have required significantly more development
work and would have increased computational costs as well, and could be a subject of a future study.

## 2.3   Experiments and analyses

First, we spun up the two models. CESM was initialized with the model state at the 507th year of the preindustrial run with the same resolution using the Community Climate System Model, version 4 (CCSM4, the model preceding CESM1.2) by Shields et al. (2012). Then the model was run for 200 years with an increased tracer time step for the deep ocean (Danabasoglu et al.,
1996). The tracer time step ($\sim$2 hours for the surface) was increased depending on the depth of the ocean: 5-times longer than at the surface for the depths from 1000 m to 2500 m, 10-times for 2500 m to 3500 m, and 20-times for 3500 m to 5500 m. Therefore, the length of the model run was equivalent to, for example, 4000 years for the very deep ocean. While most of the tracers came sufficiently close to equilibrium, the model run was not long enough for $^{14}$C that has the longer timescale of radioactive decay, and thus, we do not discuss $^{14}$C-related results in this study. By using the same acceleration method in
all CESM experiments of this study, we assumed that the influence of the model bias due to the acceleration (Danabasoglu, 2004) was mitigated as long as the differences among experiments were discussed. For this CESM spin-up, MEDUSA was not coupled but the original empirical particle-dissolution treatment of the BEC model was used instead. After the CESM spin-up, MEDUSA was initialized with a nominal uniform composition of the sediment-core-layer stack and pore water, and spun up for $10^5$ years driven by the boundary conditions (i.e., solid-particle fluxes and bottom-water chemical composition in
the deepest grid cells of POP2) derived from the spun-up CESM model state.

Following the spin-up sequence, we made two experiments. The first was a sequentially-coupled CESM-MEDUSA run for another 100 surface years with the same acceleration method for the deep ocean as described above (EXCPL). The second one was also run for 100 surface years but as a continuation of the "uncoupled" CESM spin-up run (EXORG). The latter experiment was done to examine the effect of the coupling of the process-based sediment model at millennial timescales. Again, it was
not long enough for $^{14}$C to achieve reasonable steadiness. In EXCPL, the two models communicated with each other 10 times during the 100 years; that is, the coupling interval was 10 surface years for CESM, which was equivalent to 200 years for the deepest ocean domain (i.e., deeper than 3500 m). At the end of each 10-surface-year CESM simulation, the annual mean values of the necessary variables from the last surface year were passed to MEDUSA. We ran MEDUSA for 200 years each time with a 10-year time step, and the model output at the last time step was used as input to CESM. Giving priority to the deepest ocean
domain that occupies as much as $\sim$70% of total ocean-floor area, we set the length of one MEDUSA run to 200 years that is in line with the length of the CESM integration for the deepest ocean domain.

Model performance was assessed by comparing the results to several observation-based data sets. The most straightforward benchmark quantities in the context of model–data comparison relevant for this study are the weight fractions of the solid components in the upper sediment. Here we focus on the surface sediment calcite, opal and organic carbon for which Seiter





et al. (2004) provide appropriate global gridded maps. Another important parameter is the degree of saturation of seawater with respect to calcite, defined by:

$$\Omega = \frac{[\text{Ca}^{2+}][\text{CO}_3^{2-}]}{K_{\text{calc}}} \tag{1}$$

where $K_{\text{calc}}$ denotes the solubility product of calcite. $\Omega > 1$ in waters that are supersaturated with respect to calcite; $\Omega < 1$ in

waters that are undersaturated. We use the global map of Dunne et al. (2012) as a target dataset for the seafloor calcite saturation state derived in our coupled model.

## 3    Results

### 3.1    Performance of the coupled model

First, we evaluate the performance of the ocean component of the coupled model based on the average over the last CESM run

(i.e., 10 surface years) in EXCPL. The maximum transport of the Atlantic meridional overturning circulation (AMOC) is 16.6 Sv (1 Sv = $10^6$ m$^3$/s), and the volume transport at 26.5°N (20°S) is 13.9 Sv (12.5 Sv), showing that the physical ocean state is reasonably consistent with the estimates of the modern time-mean values given by several data assimilation studies (Stammer et al., 2016). Given the physical ocean state, the BEC model is able to reproduce the observation-based estimates of various global-scale biogeochemical quantities that are relevant to this study such as the global export rates of POC and $\text{CaCO}_3$, and

their deposition rate at the ocean floor (Table 1).

To evaluate the model performance of the sediment part in the coupled model, we diagnostically obtained the weight fraction of solid components in the upper sediment from the outputs of MEDUSA. The weight fraction for calcite (Fig. 2) is of special interest considering that it is closely connected to the atmospheric $\text{CO}_2$ level variations at the glacial–interglacial timescale (e.g., Archer et al., 2000; Brovkin et al., 2007; Munhoven, 2007; Brovkin et al., 2012). The global mean weight fraction of

calcite is 21% at the last time step of the last MEDUSA run in EXCPL and 38% for the observation-based global map (Seiter et al., 2004). Although the coupled model underestimates the fraction of calcite preserved in the upper sediments, that could be partly because the global supply of calcite to the ocean floor itself is somewhat underestimated (Table 1).

We also analysed the model performance in seven geographical regions (Fig. 2c). In five regions, the magnitudes of the differences in the region-mean calcite weight fractions between EXCPL and the observation-based data are smaller than 0.15.

In particular, small model–data mismatches are realized in the North Atlantic, the western Pacific, and the Southern Ocean.

In the North Atlantic, the spatial distributions are not consistent, although the modelled region-mean weight fraction is comparable to that derived from the data. For example, the calcite weight fraction is significantly higher in the western North Atlantic in the model results than in the observation-based data by Seiter et al. (2004). A more recent observation-based data set (Dutkiewicz et al., 2015), however, indicates that the major lithology of ocean-floor sediments is calcareous matter in the

western North Atlantic, the Caribbean Sea, and the Gulf of Mexico. Although we cannot make a quantitative comparison between the newer data set and our model result because Dutkiewicz et al. (2015) does not provide the weight fraction, the model result seems to be consistent with the recent data in those regions. The discrepancy between the two data sets makes the





interpretation of the model results in the western North Atlantic elusive. On the other hand, it is also presumable that the model overestimates the supply of calcite to the sediment in the mid-west area of the North Atlantic because the modelled phosphate concentration in the surface water is higher than observed in that region (not shown), which would lead to the overestimate of biological production.

Noticeable discrepancies in the region-mean calcite weight fractions are found in the eastern South Pacific and in the Indian Ocean. The Pacific anomaly, which shows a too low modelled calcite weight fraction, is caused by too corrosive bottom water. The $\Omega$ of the bottom water in that area is lower in the model results than in the observation-based data (Fig. 3). The strongly undersaturated water having too low pH is caused by the remineralization of an anomalous amount of OM, as indicated by the too high concentration of phosphate in the deep water in this region (Fig. 4a-c) and by too much consumption of oxygen as

well (Fig. 4d-f), which are consistent with the too high weight fraction of POC in the eastern equatorial South Pacific (Fig.5a). A similar explanation is applicable to the Indian Ocean where the modelled calcite weight fraction is also lower than in the observation-based estimates; the model similarly underestimates $\Omega$ in that region, which affects the preservation of calcite in the sediments.

The simulated weight-fraction fields for POC and opal shows that they are minor components in general compared to calcite,

and that is consistent with the observation-based data (Fig. 5). Although some model–data mismatches are visible mainly in coastal areas (for POC) and in the Southern Ocean (for opal), the performance of the coupled model is remarkably better than that of the uncoupled model regarding the weight fraction of the two components (Fig. 6; see also Section 3.2).

While the model performance with regard to the calcite weight fraction may be improved to some extent by changing the model parameters of MEDUSA that govern the calcite dissolution rate, we keep the default parameter values for EXCPL in this

study, which helps to assess the model performance in a standard setting. We judge the general model performance including the reproduction of the approximate pattern of global solid weight-fraction fields to be adequate at this stage, at least for the following analyses and discussion that does not require an accurate reproduction of the observations.

## 3.2   Comparison with the uncoupled model

Although the development of the coupled model in this study has been motivated by the aim of simulating the glacial-

interglacial variations including the marine carbon cycle as an open system (Sigman and Boyle, 2000), we find that the sediment-model coupling has non-negligible influences on ocean biogeochemistry even in millennial-time-scale simulations. The most prominent effect is found in the composition of the upper sediment. For EXORG, we estimated the weight fraction of the solid components by taking the ratio of the amounts of each component and total solid matter that was excluded from the model ocean domain at the ocean floor, because the uncoupled CESM did not have explicit sediment stacks.

The weight-fraction distribution for EXORG (Fig. 6) shows that the uncoupled model behaves differently. The rough feature of the global distribution of the calcite weight fraction in EXORG is similar to that in EXCPL or the observation-based data because of the appropriate depth of the prescribed lysocline. However, EXORG underestimates the fraction mainly in the Atlantic, presumably because the spatially-constant lysocline is at shallower depths than observed in that area. On the other hand, the weight-fraction distributions for organic carbon and opal show obvious discrepancies between the model results and





the data (Fig. 6b, c). The weight fraction of organic carbon in EXORG is unreasonably higher than in the data in most regions of the global ocean and does not even approximately reproduce the observed spatial pattern. This is the case also for the opal weight fraction. The opal fraction in EXORG clearly deviates from the data, although it implies a higher fraction in the Southern Ocean and the eastern-equatorial Pacific as suggested by the observations. Those results suggest that the MEDUSA-coupling

to CESM is essential for the direct model–data comparison of the upper sediment properties.

As to the ocean state, EXORG has large-scale properties very similar to those for EXCPL; that is to say, the maximum strength of AMOC is 16.7 Sv, the global export production 8.1 GtC/yr, and the global export rain ratio of $CaCO_3$ 0.13 in EXORG, which suggests that the different treatment of the sedimentary processes does not have a remarkable effect on the overall physical and biogeochemical states of the ocean through the $pCO_2$ and dynamics of the atmosphere in our simulations.

That appears reasonable because the timescale at which the sedimentary processes alter the chemical state of the global ocean is very long ($O(10^5)$ years) considering the slow turnover rate estimated from the size of the ocean carbon reservoir and the flux exchanged with the sediments (Ciais et al., 2013).

However, the effect of the interactive coupling of MEDUSA on the local bottom-water chemistry is not negligible. The difference in $\delta^{13}C$ of DIC ($\Delta\delta^{13}C_{DIC}$; hereafter, $\Delta$ indicates the difference given by EXCPL minus EXORG) in the deepest

grid cells of the ocean model is $\sim0.2$‰ or larger in a substantial number of areas (Fig. 7a). Some of these areas correlate closely with the regions of high POC flux to the sediment (Fig. 8) or high POC weight fraction (Fig.5a): along the east coast of the equatorial Pacific, along the west coast of the Pacific, in the Arctic and Hudson Bay, for example. These regions (except for the eastern equatorial Pacific) have negative $\Delta\delta^{13}C_{DIC}$. The low $\delta^{13}C_{DIC}$ values suggest that there is a larger amount of supply of "lighter" carbon from the sediment to the seawater in such regions, which results from the remineralization of a larger

amount of OM. This is supported by the large flux of oxygen from the seawater to the sediment in the same regions (Fig. 9a) that could have been caused by a large vertical gradient of oxygen concentration, and is supported by the large DIC flux from the sediment to the seawater as well (Fig. 10). As a result, the distribution of $\Delta O_2$ and $\Delta DIC$ has a very similar spatial pattern to that of $\Delta\delta^{13}C_{DIC}$ (Fig. 7b, c). This scenario is also consistent with the $\Delta PO_4$ distribution (Fig. 7d) that is anti-correlated with that of $\Delta\delta^{13}C_{DIC}$.

A similar explanation, however, is not applicable to the eastern equatorial Pacific having positive $\Delta\delta^{13}C_{DIC}$. The seawater with larger $\delta^{13}C_{DIC}$ values suggests that there is a smaller amount of supply of lighter carbon in spite of a large flux of POM to the sediment. In that region, the amount of (oxic) remineralization in the upper sediment is limited by the reduced oxygen supply from the seawater (Fig. 9a) due to the oxygen-depleted deep water (Fig. 4d), although the model seems to underestimate the amount of oxygen available there as seen in comparisons with the corresponding observation-based fields

(Fig. 9b and Fig. 4f). The limited amount of remineralization in the model result is also consistent with the very low DIC flux from the sediment to the ocean (Fig. 10). That leads to more burial of lighter carbon (i.e., less supply of lighter carbon to the seawater), which results in the heavier $\delta^{13}C_{DIC}$ in the bottom water.

On the other hand, the remarkable dipole structure of the $\Delta\delta^{13}C_{DIC}$ field in the North Atlantic is not well correlated with the high OM-flux regions. Instead, it results from water mass displacement caused by ocean circulation changes rather than from

the direct influence of the sediment, which shows that, although the overall sediment feedback on the physical ocean states





is subtle as mentioned above, some local effects are clearly visible. The negative anomaly of $\delta^{13}C_{DIC}$ in the western North Atlantic is caused by the difference in AMOC magnitude. In EXCPL, there is a somewhat weaker penetration of northern source water into the deep low-to-mid latitude Atlantic than in EXORG (not shown). The weaker penetration means that less $^{13}$C-rich (or nutrient-depleted) surface water is transported into the deep ocean, which results in the negative $\Delta\delta^{13}C_{DIC}$. The

5   positive anomaly in the eastern North Atlantic is caused by the difference in the strength of the northward current along the African continent in the deep ocean. The current is weaker in EXCPL so that it conveys a smaller amount of $^{13}$C-depleted (or, nutrient-rich) water to the North Atlantic. As a result, the bottom water in EXCPL shows heavier $\delta^{13}C_{DIC}$ values in the eastern part of the low-latitude North Atlantic and lighter $\delta^{13}C_{DIC}$ values in the South Atlantic along the African continent. This mechanism also explains well the anomalies of the other tracers in the same regions (Fig. 7b-d).

## 4   Discussion and outlook

The most straightforward advantages of coupling CESM to MEDUSA are two-fold: First, the sediment model offers the explicit modelling of chemical and physical processes in the upper sediments, and second, modelled sediment stacks provided the climate model with sedimentary "archives". In future applications, those two advantages will facilitate a direct comparison between the climate model and (paleoceanographic) data taken from sediments, which will provide a valuable constraint on the

model from a paleoclimatological/paleoceanographic viewpoint. Otherwise, one would need to translate records obtained from sediments into corresponding variables of the ocean model, which would introduce an additional source of uncertainty to the model–data comparison. Those advantages are clearly demonstrated in the comparison of the solid weight-fraction distribution among EXCPL, EXORG, and the observation-based data. Additionally, the state of the upper sediments at a certain time has a vertical structure reflecting the "memory" of past states because the vertical mixing of the solid phase occurs by means of

bioturbation. MEDUSA has an adequate model structure including interphase biodiffusion (Munhoven, 2007) to simulate such a hysteresis effect, thereby a modelled time-series of sediment properties will be available as well.

In addition to the direct advantages, the sediment model will influence the simulated ocean biogeochemistry by providing more realistic boundary conditions. The early diagenetic processes in the upper sediments produce the chemical fluxes to the ocean, hence directly affect the chemical composition of the bottom water. The results of this study suggest that the feedback

from the upper sediments would have substantial impacts on the bottom-water chemistry even at millennial timescales. Consequently it would be worth considering carefully how to model the sediment feedback in an ocean or climate model, and a prognostic sediment model that simulates the early diagenetic processes explicitly will have an advantage, especially if it is used for a climate simulation covering different climate states and the transitions between them. In this study, the MEDUSA coupling produces $\delta^{13}C_{DIC}$ differences as much as 0.2‰ compared to the original BEC method over large areas. This result

indicates that neglecting of the sediment processes may cause a large error in the modelled chemical compositions of the bottom water. We emphasize that this error is significant in (paleo-)ocean simulations because its magnitude is close to 10% of the typical range of $\delta^{13}C_{DIC}$ values in the ocean. For example, it is comparable with the prescribed uncertainty for fitting a model to data in a paleo state estimate (Kurahashi-Nakamura et al., 2017) that takes proxy data from benthic foraminifera into





account. This will be even more important in case the other components of uncertainty have smaller contributions as implied by Breitkreuz et al. (2018).

As a future application of the coupled model, we aim at investigating the role of sedimentary diagenesis in the climate changes at glacial–interglacial timescales. In this context, one of the future tasks will be a simulation of the evolution of the

atmospheric carbon dioxide concentration ($pCO_2$) as recorded in Antarctic ice cores (Berner et al., 1980; Barnola et al., 1987; Augustin et al., 2004). A simulation of the history of $pCO_2$ at such timescales is one of the crucial challenges of climate science, and it is widely considered that the ocean could have played a key role in the $pCO_2$ variations because of its dominant size as a carbon reservoir (e.g., Sigman and Boyle, 2000; Ciais et al., 2013). The budget of $CaCO_3$ in the global ocean, that is, the balance of the $CaCO_3$ inflow by weathering on land and the outflow by sedimentary burial, would have had a substantial

effect on the distribution of total carbon to the ocean and atmosphere, hence $pCO_2$, by changing the acidity or basicity of the entire ocean (e.g., Archer et al., 2000; Matsumoto et al., 2002; Brovkin et al., 2007; Munhoven, 2007; Chikamoto et al., 2008; Boudreau et al., 2010, 2018). It is highly important, therefore, to properly simulate the preservation or dissolution of $CaCO_3$ in ocean-floor sediments in order to handle the mechanism quantitatively. The relatively good agreement of the calcite weight fraction between EXORG and the observation-based field suggests that calcite preservation depends mainly on water depth and

implies that the "fixed-lysocline" method might be practical for a given ocean state as long as an appropriate threshold depth can be prescribed. However, generally the depth of lysocline is not constant but depends on the ambient seawater chemistry. This indicates that the fixed depth optimized for one ocean state is not necessarily suitable for another ocean state. Therefore, large-scale and long-term climate change studies will certainly require a dynamical sediment diagenesis model. More specifically, MEDUSA will provide CESM with the crucial ability to simulate the feedback between the $CaCO_3$ budget and the global

ocean chemistry that is often referred to as carbonate compensation (e.g., Broecker and Peng, 1987; Archer et al., 2000).

## 5   Conclusions

We coupled a dynamical model of early diagenesis in ocean sediments (MEDUSA) to the ocean component including a biogeochemical module of an advanced comprehensive climate model (CESM1.2). A simulation for the modern climate state demonstrated that the coupled CESM-MEDUSA model is able to approximately simulate the observed global patterns of solid

composition in the upper sediments.

The comparison between the coupled and uncoupled models shows that the coupling of MEDUSA only has minor effects on the bulk properties of the global ocean in millennial-timescale climate simulations as expected from the characteristic timescale of sedimentary processes. This study, however, reveals that the sediment-model coupling is significant in two aspects even at such a timescale. First, the simulated sediments provides an additional measure of model performance, and the observation-

based global distributions of sediment properties are much better reproduced by CESM coupled to MEDUSA than by the uncoupled CESM. Secondly, some immediate effects of the sediment-model coupling are found in the chemical composition of the bottom water. The difference in the chemical composition of the bottom water between the MEDUSA-coupled model and the uncoupled model is large in the regions of high POC flux to the sediment, which suggests that it would be important



to simulate the remineralization of POC in the upper sediments appropriately depending on the bottom-water chemical composition (e.g., oxygen availability). Additionally, the different treatments of the sediment processes can result in some visible displacement of the water masses in the deep ocean, which causes the different distributions of chemical tracers.

The MEDUSA coupling will yield another remarkable advantage over the original model with regard to the $CaCO_3$ dynamics. For long-term climate simulations including the global carbon cycle, the dynamical treatment of the $CaCO_3$ dissolution or preservation in the upper sediments will be essential. We consider the MEDUSA–CESM coupled model as a powerful tool to explore the climate dynamics at glacial-interglacial timescales that will give new insights into the feedback between the sediment processes and the global climate.

*Code and data availability.* The newly developed model source codes to tailor CESM1.2 and MEDUSA (version 359 or newer) for the coupling and the routines to make input files for either model from output files of the other are available in https://doi.pangaea.de/10.1594/PANGAEA.905821.

*Author contributions.* TK-N developed the model code for the coupling with input from AP and GM. TK-N and AP designed the experiments, and TK-N carried them out. TK-N interpreted and discussed the results with contributions from all co-authors. AP, UM, and MS conceptualized the overarching research goal and acquired the financial support leading to this publication. TK-N prepared the manuscript with contributions from all co-authors.

*Competing interests.* This study has no competing interests.

*Acknowledgements.* This research was funded by the project PalMod (www.palmod.de; FKZ: 01LP1505D) within the framework of Research for Sustainable Development (FONA, http://fona.de) by the German Federal Ministry for Education and Research (BMBF). GM is a Research Associate with the Belgian Fund for Scientific Research-FNRS.





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



**Table 1.** Comparison of globally-integrated biogeochemical quantities of this study with previous estimates available from observations. For EXCPL, fluxes to allow consistent comparison with the previous estimates were calculated from the time averages over the last CESM run (10 surface years).

| | This study (EXCPL) | Previous estimates |
|---|---|---|
| **Atmospheric $p\mathrm{CO}_2$ (ppm)** | | |
| Global mean | 276.9 | 285.2 (Etheridge et al., 1996) |
| **Net primary production (GtC/y)** | | |
| | 59.4 | 48.5 (Field et al., 1998) |
| | | 49 – 60 (Carr et al., 2006) |
| | | 54 (Dunne et al., 2007) |
| | | 48.2 (Laws et al., 2011) |
| | | 55 (Ma et al., 2014) |
| **POC (Gt/y)** | | |
| Export at 100 m | 7.2 | 9.6 ± 3.6 (Dunne et al., 2007) |
| | | 9.2 – 13.2 (Laws et al., 2011) |
| | | 4.0 (Henson et al., 2012) |
| | | 5.7 (Siegel et al., 2014) |
| Flux to ocean floor below 1000 m between 60°N and 60°S | 0.33 | 0.13 (Dunne et al., 2007) |
| **CaCO₃ (GtC/y)** | | |
| Export at 100 m | 0.95 | 1.1 ± 0.3 (Lee, 2001) |
| | | 0.5 – 4.7 (Berelson et al., 2007) |
| | | 0.52 ± 0.15 (Dunne et al., 2007) |
| | | 0.72 – 1.05 (Battaglia et al., 2016) |
| Flux to ocean floor below 2000 m | 0.32 | 0.5 ± 0.3 (Berelson et al., 2007) |
| **Opal (Tmol/y)** | | |
| Export at 100 m | 88.6 | 120 (Tréguer et al., 1995) |
| | | 70 – 100 (Gnanadesikan, 1999) |
| | | 101 ± 35 (Dunne et al., 2007) |
| | | 105 (Tréguer and De La Rocha, 2013) |
| Flux to ocean floor | 46.4 | 29.1 (Tréguer et al., 1995) |
| | | 22.1 (Dunne et al., 2007) |
| | | 78.8 (Tréguer and De La Rocha, 2013) |





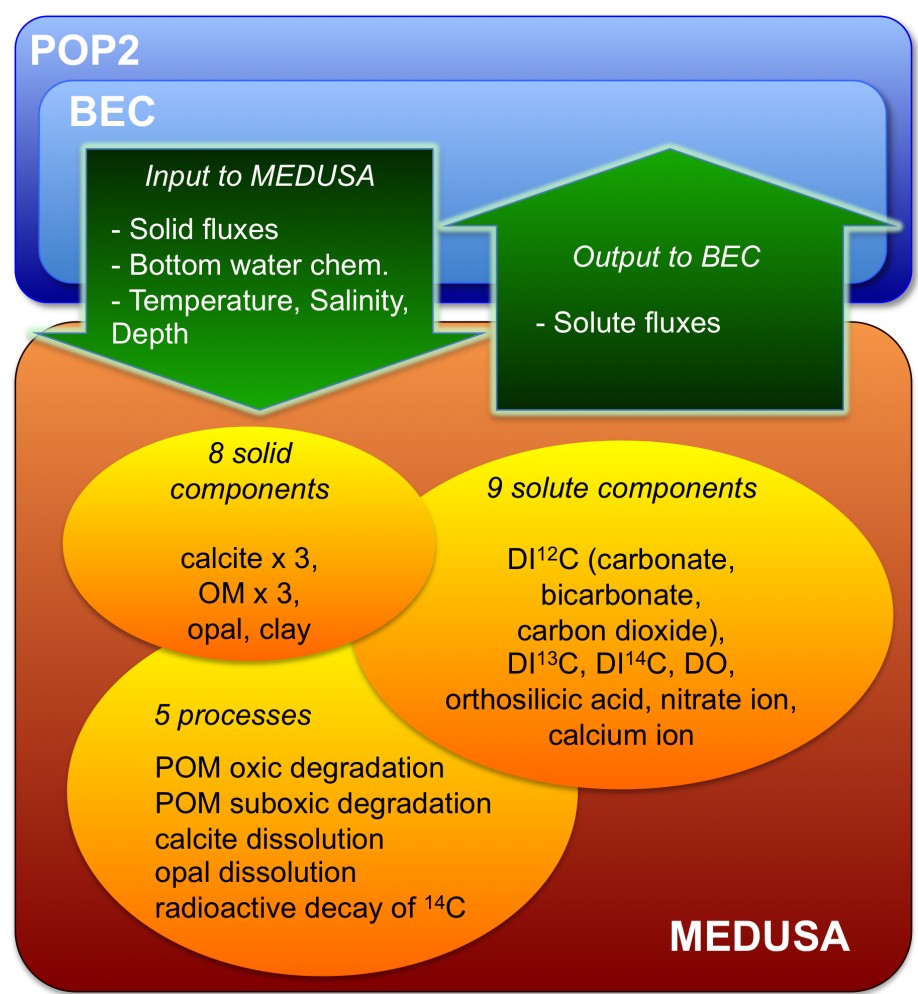

**Figure 1.** A schematic illustration of this study's coupling scheme. In the list of chemical species, "D" stands for "dissolved" and "I" for "inorganic"; for example, DO means dissolved oxygen and DIC dissolved inorganic carbon. OM stands for organic matter. Each of OM and calcite components had three categories (for $^{12}$C, $^{13}$C and $^{14}$C), and they were treated as separate tracers.





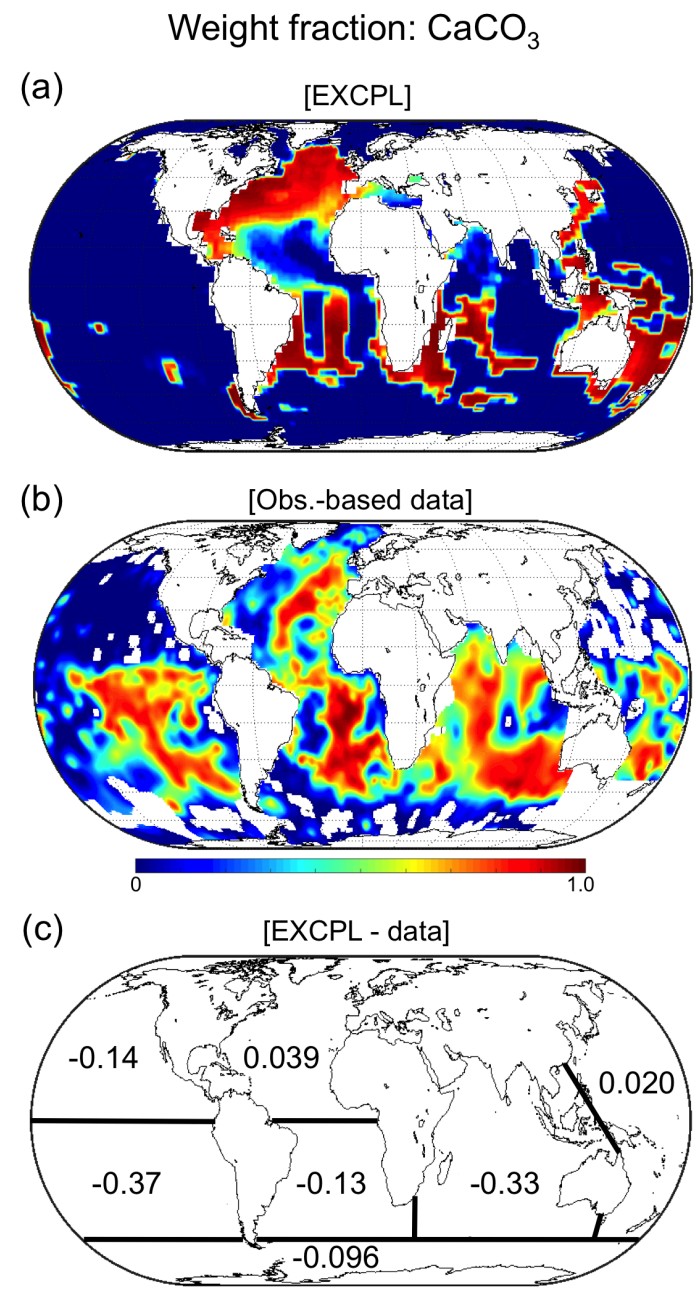

**Figure 2.** The weight fraction of the $CaCO_3$ component in the upper sediments: (a) the model state at the last time step of the last MEDUSA run in EXCPL and (b) the gridded map derived from observations (Seiter et al., 2004). The characteristic timescale of sediment-stack evolution is much longer than 10 years, so that the weight fraction at the last time step sufficiently represents the model state. (c) The breakdown of the model–data discrepancies into seven regions. The differences of the regional mean values for each domain are shown (EXCPL minus data).

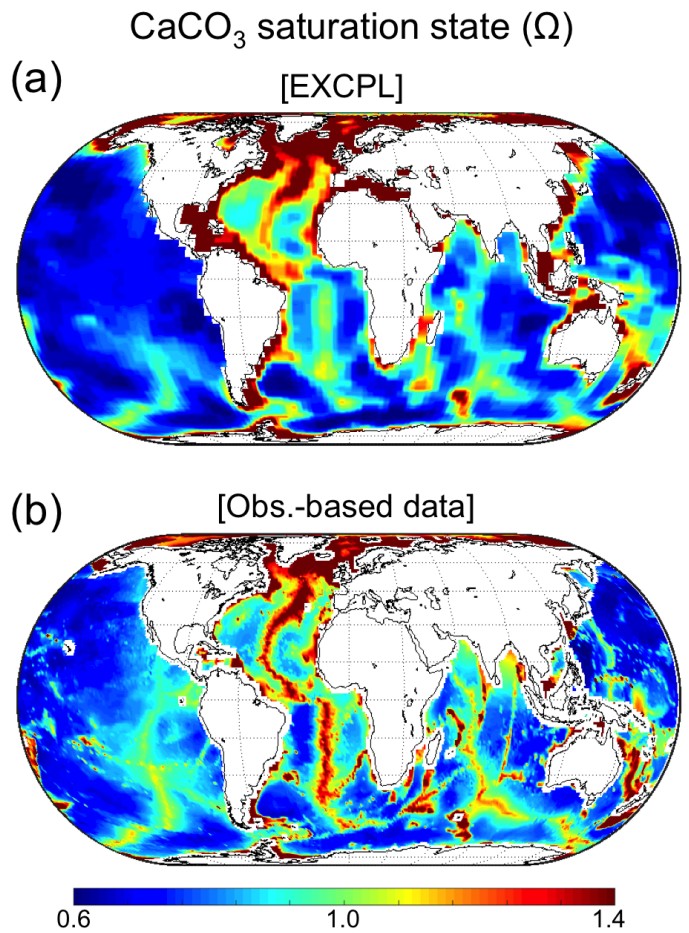

**Figure 3.** Saturation state for calcite of the bottom seawater ($\Omega$). (a) The model state in the deepest grid cells averaged over the last CESM run (10 surface years) in EXCPL, and (b) observation-based estimates by Dunne et al. (2012).





**Figure 4.** Distribution of marine biogeochemical tracers: (a-c) dissolved inorganic phosphate at the depth of 3000 m, (d-f) dissolved oxygen at 3000 m, and (g-i) dissolved inorganic silicate at 10 m. The left column shows the time averages over the last CESM run (10 surface years) in EXCPL, the centre shows observation-based data (GLODAPv2; Lauvset et al., 2016), and the right anomaly given by model results minus the data.



## [EXCPL]

## [Obs.-based data]

(a) weight fraction: org. C

(b) weight fraction: org. C

0                    0.03

(c) weight fraction: opal

(d) weight fraction: opal

0                    0.5

**Figure 5.** The weight fraction of organic carbon and opal in the upper sediments. The left column shows the model states at the last time step of the last MEDUSA run in EXCPL, and the right column show the respective gridded map derived from observations (Seiter et al., 2004).

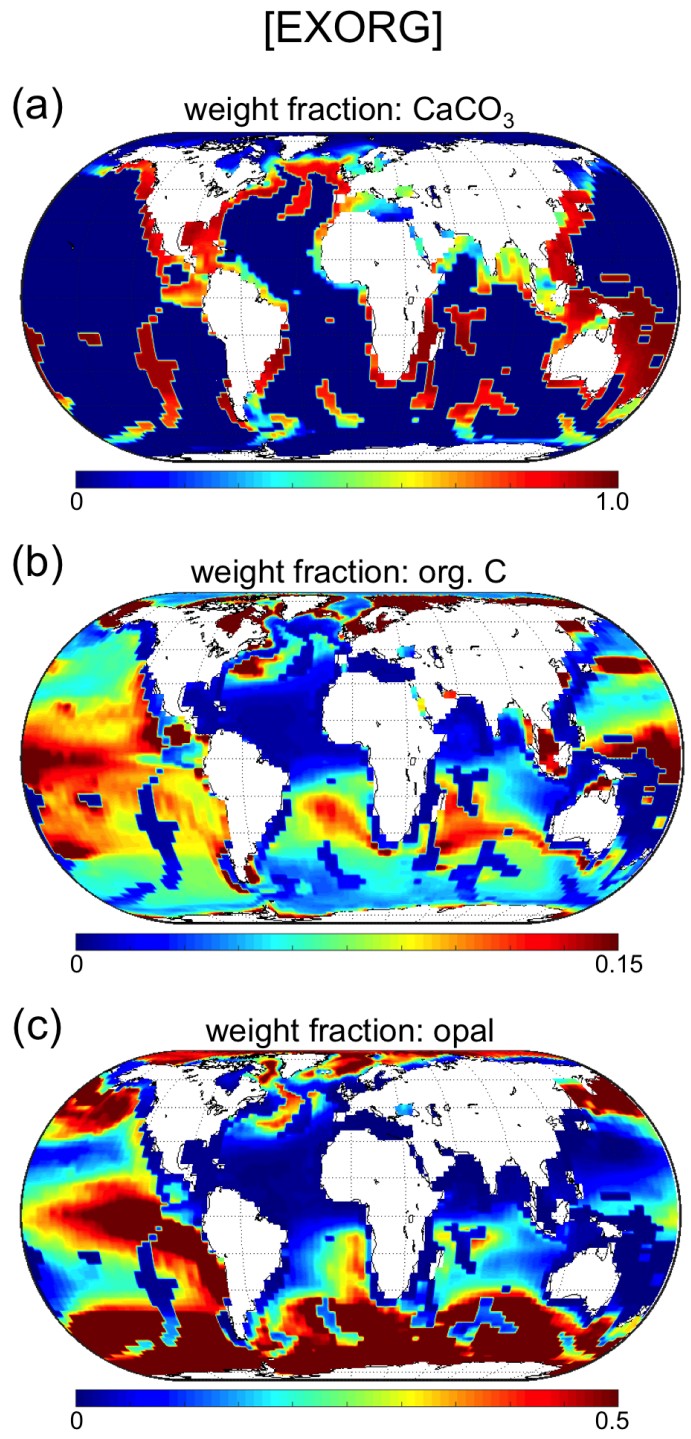

**Figure 6.** The weight fraction of three solid components in the upper sediments for EXORG. We estimated the weight fraction by taking the ratio of the amount of solid matter that was excluded from the model ocean domain at the ocean floor based on the time averages over the last 10 surface years of the CESM run. Note that the contour scale for organic carbon is different from that for EXCPL in Fig. 5.

## [EXCPL – EXORG]

(a)   Δ δ¹³C

(b)   Δ O₂

(c)   Δ DIC

(d)   Δ PO₄

**Figure 7.** The difference in the chemical composition of the deepest grid cells between EXCPL and EXORG that was obtained from the time averages over the last 10 surface years of the CESM run in each experiment. Δ indicates an anomaly given by EXCPL minus EXORG.





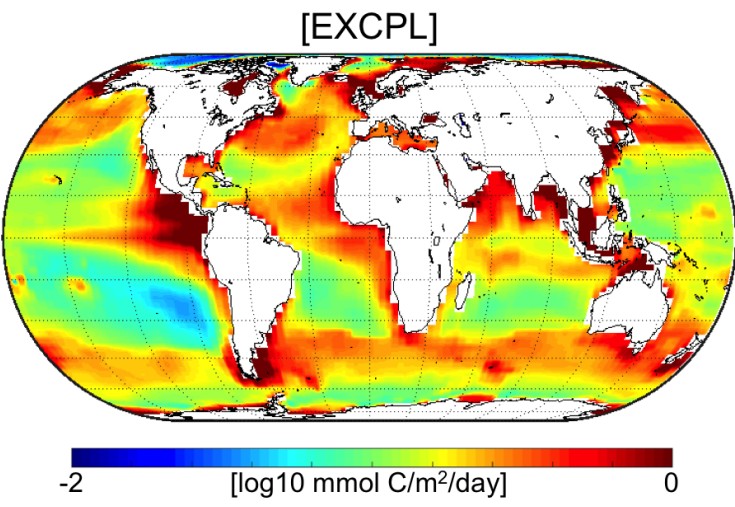

**Figure 8.** The flux of particulate organic carbon at the top of the sediment. The time averages over the last CESM run (10 surface years) in EXCPL are shown. The same scale as that in Fig. 5a of Dunne et al. (2007) is used to facilitate comparison.

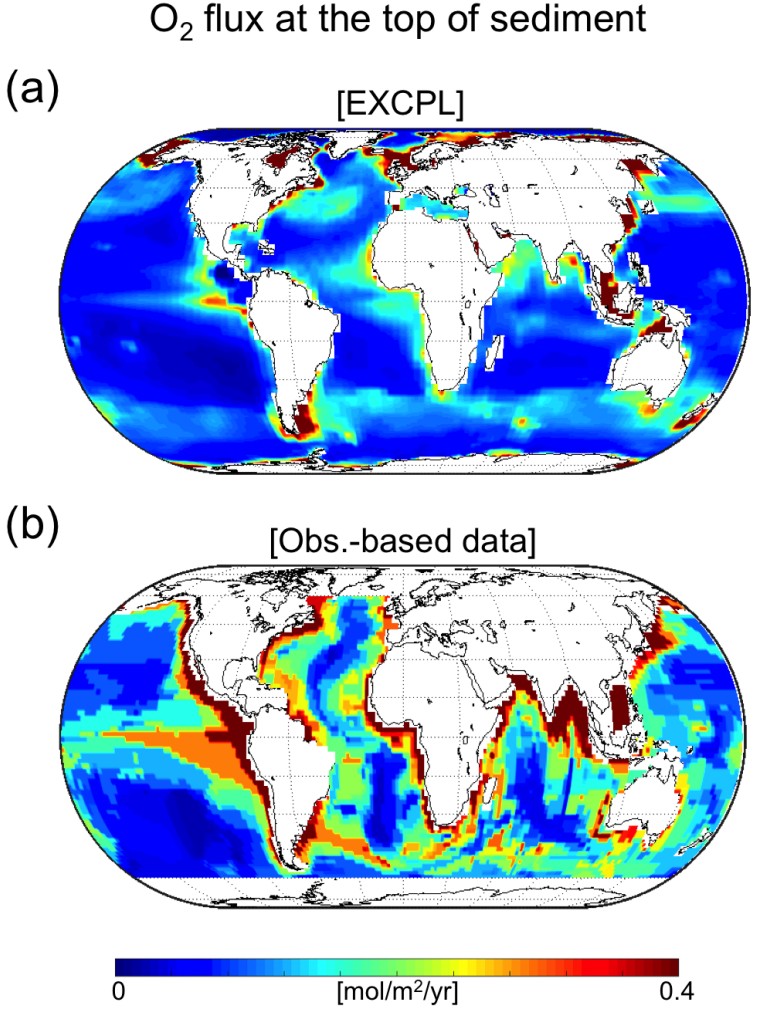

**Figure 9.** Oxygen fluxes at the water–sediment boundary (a) at the last time step of the last MEDUSA run in EXCPL and (b) observation-based estimate (Jahnke, 1996). The sign is downward positive; that is to say, positive values correspond to fluxes from ocean to sediments.





**Figure 10.** DIC fluxes at the water–sediment boundary at the last time step of the last MEDUSA run in EXCPL. The sign is downward positive; that is to say, positive values correspond to fluxes from ocean to sediments.