# Peer review of "Coupling of a sediment diagenesis model (MEDUSA) and an Earth system model (CESM1.2): a contribution toward enhanced marine biogeochemical modelling and long-term climate simulations"

_Geoscientific Model Development, 2019_

## Referee Comment (RC1) · Anonymous Referee #1 · 3 Nov 2019

This paper developed the coupled system of sediment processes with the earth system using a sediment model (MESUDA) and a state-of-the-art fully coupled atmosphere-ocean-sea ice-land model (CESM1.2), and then diagnosed the sediment composition contents under the modern ocean conditions. Surprisingly, the comparison between the coupled and the uncoupled system between ocean and sediment processes shows a substantial difference in stable carbon isotope concentrations in some regions of coastal regions and the equatorial Pacific, which suggests that the sediment interaction

with bottom ocean waters affects the distributions of stable carbon isotopes as well as other biogeochemical compositions. This result indicates the importance of the coupled system between ocean and sediment for the biogeochemical simulation over millennial time scales. In addition, while the previous model studies applied for the coupling of a sediment model with earth system models of intermediate complexity, this study couples the sediment model with the state-of-the-art comprehensive climate model. This development is technically advanced. I recommend for publication within minor revision. My comments and suggestions regarding the evaluation of the model's capability are as follows.

Major comments

(1) Large differences in sediment contents between two experiments

The paper shows the diagnosed sediment contents of opal and organic carbon are very different in the coupled (EXCPL) and uncoupled (EXOGR) ocean-sediment system (Figs. 5a,c and 6b–c). Why is the difference so large between the two experiments? I expect that the sediment contents should be relatively similar in two experiments, as shown by the relative similarity in CaCO3 sediments (Figs. 2a and 6a), because the sedimentation feedback seems to be small in the broad ocean expect the North Atlantic (Fig 7).

I wonder whether the total concentrations of silicate and nutrient are conserved in EX-CPL. It may be helpful to provide a table that presents the global integrated deposition fluxes of opal, CaCO3 and OC and the global integrated concentrations of DIC, ALK, SiO3 or PO4 in two experiments. I also recommend to add the description how the model treats the riverine inflow and sediment outflow fluxes in Section 2.2. That information is key to understand the experimental design of EXCPL. For example, we can understand whether EXCPL is designed for an open or closed system in the atmosphere, ocean and sediment reservoirs.

(2) The impact of all dissolution of CaCO3 below 3300m depth

The BEC model is coordinated by all dissolution of particulate CaCO3 in the ocean below 3300 m depth, which probably causes less calcite preservation particularly in the Pacific and Indian Ocean. It would be helpful to discuss the impact of this "fixed lysocline depth" setting to the model performance and behaviors in more details. Does this setting affect excess accumulation of organic matter in the equatorial Pacific? I suspect that less CaCO3 burial maybe cause slower sedimentation rate, which may expose OC at the upper sediments on longer timescale and thus accelerate the decomposition of OC in sediments.

(3) What is the difference of sediment coupling with the state-of-the-art earth system model with previous studies with intermediate complexity models?

I think it is helpful to discuss the advantage using the state-of-the-art earth system model. What is a large difference in simulations between CESM-MESUDA and for example, GENIE? What does this development help our better understanding?

Minor comments:

Table1: This table is very good and informative to provide the model's capability from the model-data comparison. It may be also helpful to add the deposition and burial fluxes, as described above in my comment (1).

Page 3 L31–33: This description is unclear. Does it mean that all burial fluxes return to the bottom water as dissolved properties? Please rewrite the description.

Page 8 L5–6: This sentence is also unclear. Do you want to say that the ocean-sediment coupling is important to simulate the water properties? Please rephrase it to present your argument more clearly.

Page 9 Line29: "over large areas" maybe mislead readers. In this paper, the sediment feedback is apparent in some regions, such as along the east coast of the equatorial Pacific, along the west coast of the Pacific, in the Arctic and Hudson Bay. Rather, the large difference in d13C in the North Atlantic arises from the model's bias in relation

to AMOC or ocean mixing variability, which should be excluded from the sediment contributions to the bottom-water properties.

Page 10 L29: provides –> provide

---

## Short Comment (SC1) · 15 Nov 2019

Dear authors,

in my role as Executive editor of GMD, I would like to bring to your attention our Editorial version 1.1:

http://www.geosci-model-dev.net/8/3487/2015/gmd-8-3487-2015.html

[Figure]

This highlights some requirements of papers published in GMD, which is also available on the GMD website in the 'Manuscript Types' section:

http://www.geoscientific-model-development.net/submission/manuscript_types.html

In particular, please note that for your paper, the following requirements have not been met in the Discussions paper:

- "The main paper must give the model name and version number (or other unique identifier) in the title."

Please add a version number of MEDUSA in the title of your article in your revised submission to GMD. Additionally, please note, that the link in the code availability section is not correct / does not fully appear in the document. Please post as fast as possible the correct link in the discussion forum and change this in the final document!

Yours,

Astrid Kerkweg

---

## Author Comment (AC1) · 15 Nov 2019

Dear Astrid Kerkweg,

Thank you very much for noticing the problems and for the comments.

* About the version number of MEDUSA

We will add a version number to the title in the next version of the manuscript.

[Figure]

\* About the link

Although the link in the code availability section of the current version of the manuscript works properly in my environment, here we specify the location of the database again.

https://doi.pangaea.de/10.1594/PANGAEA.905821

As to the appearance in the manuscript, we will arrange it carefully in the next (or, final) version of the manuscript.

Best regards,

Takasumi Kurahashi-Nakamura
* * *

---

## Referee Comment (RC2) · Anonymous Referee #2 · 29 Nov 2019

General comments

I like the paper. It has a clear objective and is nicely written. I support publishing it, but just have a few suggestions how it could still be improved.

Specific comments

Page 1 Line 16: I consider this sentence as a bit unclear or vague. Please state more

explicitly which models might show a bias, and which results of those models might therefore be less reliable than previously thought.

Page 2 Line 24-25: "Up to the present, no fully-coupled comprehensive climate model hasbeen coupled with a sediment diagenesis model for longer time-scale applications (e.g., the glacial-interglacial variations)." -> Has this approach been used on shorter (e.g. centennial) time scales? Can you give examples, and how does your approach differ from them?

Page 5 Line 24-31: Can you show a figure, possibly in the online supplement, that proves that your time step was sufficiently small and your integration period sufficiently long to show something like a convergence of the sediment-water fluxes in the end (for all but the 14C of course)?

Page 7 Line 3-4: "which would lead to the overestimate of biological production" -> "which would lead to an overestimation of biological production"?

Page 9 Line 15-17: "Otherwise, one would need to translate records obtained from sediments into corresponding variables of the ocean model, which would introduce an additional source of uncertainty to the model–data comparison." -> You have an opposite translation by the MEDUSA model: ocean model variables are translated to sediment records. Why is this less uncertain than the other way around?

Table 1: Would it make sense to add a third column for the values in the EXORG run?

Figure 1: Why are the state variables only listed for the MEDUSA model and not for the BEC model? Probably the list of processes might be too long, but at least the state variables would give an indication of the model complexity for those not familiar with BEC.

Figure 6: Having this figure separate from Fig. 5 and using changed color scales makes the comparison very hard. And the improved behaviour of the model using the coupling is the main point of your manuscript. If you think the subfigures become too

small if you put all three weight fractions for EXORG, EXCPL and OBS into one figure, you might consider one figure for each weight fraction but then containing EXORG, EXCPL and OBS?

---

## Author Comment (AC3) · 10 Jan 2020

Reply to the Referee #2

We would like to thank the reviewer for the thoughtful and constructive comments.

Q1: Page 1 Line 16: I consider this sentence as a bit unclear or vague. Please state more explicitly which models might show a bias, and which results of those models might therefore be less reliable than previously thought.

A1: First, we will replace the word 'biases' with 'uncertainty' since we intended to address the general shortcomings of such models. Second, we will add the following sentence to the abstract. "For example, an ocean model that does not treat sedimentary processes depending on the chemical composition of the ambient water can overestimate the amount of remineralization of organic matter in the upper sediment under an anoxic environment, which would lead to lighter $\delta 13C_{DIC}$ in the bottom water."

Q2: Page 2 Line 24-25: "Up to the present, no fully-coupled comprehensive climate model has been coupled with a sediment diagenesis model for longer time-scale applications (e.g., the glacial-interglacial variations)." -> Has this approach been used on shorter (e.g. centennial) time scales? Can you give examples, and how does your approach differ from them?

A2: We will revise the sentence as follows: "To our knowledge, a fully-coupled comprehensive climate model including a sediment diagenesis model has been applied to millennial time scales only (Jungclaus et al., 2010). Here we aim at applying such a model to time scales of tens of millennia, having glacial-interglacial variations in mind."

Q3: Page 5 Line 24-31: Can you show a figure, possibly in the online supplement, that proves that your time step was sufficiently small and your integration period sufficiently long to show something like a convergence of the sediment-water fluxes in the end (for all but the 14C of course)?

A3: In terms of DIC flux back to the ocean, the difference between a run with the original time step and a run with only 1/10 of the original time step is smaller than 0.5% for most grid cells. We will add figures to show this in the supplementary material. Time steps shorter than 1 year do not make sense because the input from CESM is annually averaged.

It should be noted that the CESM-MEDUSA coupled simulation (EXCPL) is not a "steady-state" run but a "transient" run where the model state evolves. Therefore, the length of the MEDUSA runs is not determined by the convergence of a model state but by the coupling interval.

Q4: Page 7 Line 3-4: "which would lead to the overestimate of biological production" -> "which would lead to an overestimation of biological production"?

A4: The sentence will be corrected.

Q5: Page 9 Line 15-17: "Otherwise, one would need to translate records obtained from sediments into corresponding variables of the ocean model, which would introduce an additional source of uncertainty to the model--data comparison." -> You have an opposite translation by the MEDUSA model: ocean model variables are translated to sediment records. Why is this less uncertain than the other way around?

A5: The "forward modeling" by MEDUSA is beneficial because it provides a process-based translation rather than an empirical translation that would be inevitable without such a sediment model. Therefore, the point is not the direction of translation but the way of translation. We will revise the sentence as follows. "Otherwise, one would need to translate records obtained from sediments in an empirical way to corresponding variables of the ocean model, which would introduce an additional source of uncertainty to the model-data comparison."

Q6: Table 1: Would it make sense to add a third column for the values in the EXORG run?

A6: We will add the following two tables to compare EXCPL and EXORG in Section 3.2 to show that the two experiments are comparable in general in terms of globally-integrated quantities (except for the burial flux). In Section 3.1, we would rather focus on the comparison between EXCPL and observations as in the current version of the manuscript.

**Table 1**. Globally-integrated annual mean deposition flux of particulate matter to the sediment and their burial flux (in parentheses) at the end of EXCPL and EXORG.

|                   | EXCPL   | EXORG  |
| ----------------- | ------- | ------ |
| POC (GtC/y)       | 0.57    | 0.51   |
|                   | (0.091) | (0.12) |
| $CaCO_3$ (GtC/y)  | 0.39    | 0.38   |
|                   | (0.082) | (0.14) |
| Opal (Tmol/y)     | 46      | 45     |
|                   | (0.72)  | (3.4)  |

**Table 2**. Total inventories in the global ocean of DIC, ALK, and PO$_4$ in EXCPL and EXORG. Values averaged over the last CESM run (10 surface years) are shown.

|  | EXCPL | EXORG |
|---|---|---|
| DIC (GtC) | $3.660 \times 10^4$ | $3.657 \times 10^4$ |
| ALK (Peq) | $3.201 \times 10^3$ | $3.201 \times 10^3$ |
| PO$_4$ (Pmol) | 2.948 | 2.923 |

Q7: Figure 1: Why are the state variables only listed for the MEDUSA model and not for the BEC model? Probably the list of processes might be too long, but at least the state variables would give an indication of the model complexity for those not familiar with BEC.

A7: We will update the figure as follows.

[Figure]

Q8: Figure 6: Having this figure separate from Fig. 5 and using changed color scales makes the comparison very hard. And the improved behaviour of the model using the coupling is the main point of your manuscript. If you think the subfigures become too small if you put all three weight fractions for EXORG, EXCPL and OBS into one figure, you might consider one figure for each weight fraction but then containing EXORG, EXCPL and OBS?

A8: We will re-arrange the figures according to the reviewer's suggestion.

---

## Author Response (AR1)

Reply to the Referee #1

We would like to thank the reviewer for the thoughtful and constructive comments.

<< About the major comments >>

Q1a: Large differences in sediment contents between two experiments

The paper shows the diagnosed sediment contents of opal and organic carbon are very different in the coupled (EXCPL) and uncoupled (EXOGR) ocean-sediment system (Figs. 5a,c and 6b--c). Why is the difference so large between the two experiments? I expect that the sediment contents should be relatively similar in two experiments, as shown by the relative similarity in CaCO3 sediments (Figs. 2a and 6a), because the sedimentation feedback seems to be small in the broad ocean expect the North Atlantic (Fig 7).

A1a: Burial ratios (the ratios of burial amount to the flux to the ocean bottom) of OM and opal calculated by MEDUSA in EXCPL are remarkably different from those given by the highly simplified parameterization in the original CESM (Fig. r1). In particular, the ratios in EXCPL are significantly lower in low-flux locations, which means that the difference will be larger in the open ocean. Depending on whether OM or opal forms the major part of the total particulate flux (e.g., opal in the Southern Ocean), the difference in burial ratios will lead to substantial discrepancies in terms of the weight fraction.

In this regard, we added a new figure (Fig. 7 in the revised manuscript) and the following sentences to Section 3.2 in the manuscript: "The noticeable differences in the weight fractions of OC and opal between EXCPL and EXORG are mostly caused by the different degrees of preservation of those two species in the upper sediment. Burial ratios (the ratios of burial amount to the flux to the ocean bottom) of OM and opal calculated by MEDUSA in EXCPL are remarkably different from those given by the highly simplified parameterization in the original CESM (Fig. 7). In particular, the ratios in EXCPL are significantly lower in low-flux locations, which means that the difference will be larger in the open ocean. Depending on whether OM or opal forms the major part of the total particulate flux (e.g., opal in the Southern Ocean), the difference in burial ratios will lead to substantial discrepancies in terms of the weight fraction."

[Figure]

**Figure r1**. Sediment burial ratios versus the particulate flux to the ocean floor for (a) OC and (b) opal. The dots show the ratio at each grid cell obtained in the last MEDUSA run for EXCPL. The solid lines indicate those given by the parameterized models in the original CESM(BEC) based on Dunne et al. (2007) for OC and Ragueneau et al. (2000) for opal.

Q1b: I wonder whether the total concentrations of silicate and nutrient are conserved in EXCPL. It may be helpful to provide a table that presents the global integrated deposition fluxes of opal, CaCO3 and OC and the global integrated concentrations of DIC, ALK, SiO3 or PO4 in two experiments.

A1b: We added the following two tables (Tables 2 and 3 in the revised manuscript) including those globally integrated quantities to Section 3.2 to show that the two experiments are comparable in terms of globally-integrated quantities except for the burial flux.

**Table r1**. Globally-integrated annual mean deposition flux of particulate matter to the sediment and their burial flux (in parentheses) at the end of EXCPL and EXORG.

|  | EXCPL | EXORG |
|---|---|---|
| POC (GtC/y) | 0.57 (0.091) | 0.51 (0.12) |
| CaCO$_3$ (GtC/y) | 0.39 (0.082) | 0.38 (0.14) |
| Opal (Tmol/y) | 46 (0.72) | 45 (3.4) |

**Table r2**. Total inventories in the global ocean of DIC, ALK, and PO$_4$ in EXCPL and EXORG. Values averaged over the last CESM run (10 surface years) are shown.

|  | EXCPL | EXORG |
|---|---|---|
| DIC (GtC) | $3.660 \times 10^4$ | $3.657 \times 10^4$ |
| ALK (Peq) | $3.201 \times 10^3$ | $3.201 \times 10^3$ |
| PO$_4$ (Pmol) | 2.948 | 2.923 |

Also, we added the following sentence to to Section 3.2: "The two experiments are also comparable in terms of other globally-integrated quantities such as the deposition fluxes of particulate matter and the total inventories of biogeochemical tracers (Tables 2 and 3)."

Q1c: I also recommend to add the description how the model treats the riverine inflow and sediment outflow fluxes in Section 2.2. That information is key to understand the experimental design of EXCPL. For example, we can understand whether EXCPL is designed for an open or closed system in the atmosphere, ocean and sediment reservoirs.

A1c: Both EXCPL and EXORG are designed as an "open" system. Both experiments have a common riverine-inflow field corresponding to the modern nutrient exports based on Seitzinger et al. (2010) and Mayorga et al. (2010). On the other hand, the net flux of matter through the lower boundary of the ocean domain is calculated by MEDUSA in EXCPL and by the parameterized burial treatment of BEC in EXORG.

We added a description regarding the open-system configuration to Section 2.3 rather than 2.2 because it is a common framework to EXCPL and EXORG as follows: "Both EXCPL and EXORG are designed as an "open" system. Both experiments have a common riverine-inflow field corresponding to the modern river nutrient exports based on Seitzinger et al. (2010) and Mayorga et al. (2010). On the other hand, the net flux of matter through the lower boundary of the ocean domain is calculated by MEDUSA in EXCPL and by the parameterized burial treatment of BEC in EXORG."

Q2: The impact of all dissolution of CaCO3 below 3300m depth
The BEC model is coordinated by all dissolution of particulate CaCO3 in the ocean below 3300 m depth, which probably causes less calcite preservation particularly in the Pacific and Indian Ocean. It would be

helpful to discuss the impact of this "fixed lysocline depth" setting to the model performance and behaviors in more details. Does this setting affect excess accumulation of organic matter in the equatorial Pacific? I suspect that less CaCO3 burial maybe cause slower sedimentation rate, which may expose OC at the upper sediments on longer timescale and thus accelerate the decomposition of OC in sediments.

A2:

The difference between the prescribed fixed-depth of $CaCO_3$ dissolution and the actual depth of lysocline is larger in the Atlantic Ocean than in the Pacific and Indian Oceans. In EXORG, therefore, the influence of the fixed depth on the $CaCO_3$ weight fraction is more noticeable in the Atlantic when compared to the observation-based data, as mentioned in the manuscript (p.7, l.32)

As to the excess accumulation of OM in the equatorial Pacific in EXORG, we find that the effect of the simplified OM dissolution scheme dominates over the reduced burial of $CaCO_3$ (see our answer A1a to question Q1a) because such an excess accumulation of OM is not observed in EXCPL where there is hardly any $CaCO_3$ burial in that region as in EXORG.

However, the reviewer's argument applies to EXCPL and explains the underestimation of the OC weight fraction in the eastern South Pacific (around 110°W, 25°S) and the correlation between the patterns in the OC and $CaCO_3$ weight fractions in that region. We added the following sentence to the 6th paragraph of Section 3.1: "In some regions, for example in the eastern South Pacific (around $110^{\circ}$W, $25^{\circ}$S), the simulated OC weight fraction is lower than the observed OC fraction. This correlates with the underestimation of the calcite weight fraction (Fig. 2), which implies that less calcite burial may cause a slower sedimentation rate, leading to a longer exposure of OC to the pore water in the upper sediment and thus facilitating its respiration."

Q3: What is the difference of sediment coupling with the state-of-the-art earth system model with previous studies with intermediate complexity models? I think it is helpful to discuss the advantage using the state-of-the-art earth system model. What is a large difference in simulations between CESM-MESUDA and for example, GENIE? What does this development help our better understanding?

A3:

We consider that the advantage of using state-of-the-art comprehensive models over using Earth system models of intermediate complexity (EMICs) is (at least) threefold:

First, EMICs typically use more empirical parameterizations than process-based representations of physical (and other) phenomena in their model components to realize a more efficient computation. For many

EMICs, this applies in particular to the atmosphere component. Such model representations cannot properly capture the feedback from variations in model input if it is beyond the range of the underlying empirical relationship. From this viewpoint, comprehensive models would be more advantageous to simulate the response of the atmosphere or the ocean to the variation in the sediment component in a long-term transient "paleo" simulation that explores climate states very different from the present-day.

Second, the ocean component of some EMICs is of lower dimension and/or coarser spatial resolution. For example, the ocean component of CLIMBER-2 is based on zonally-averaged equations for three ocean basins with a meridional resolution of 5°, while the ocean component of cGENIE is three-dimensional but of a similar coarse horizontal resolution and using simplified ("frictional") physics. Using primitive equations in the atmosphere and ocean combined with a higher spatial resolution is a clear advantage in comparing model results to local observations because it reduces the uncertainty introduced by the mapping, averaging or interpolation of either model output or data.

Third, as an indirect merit, it enables us to evaluate the performance of comprehensive CMIP5-level climate models with respect to additional observational data sets from a new archive (i.e., ocean sediments), which is a significant benefit, considering that the assessment of model performance is a crucial task in the global-climate-projection context (e.g., Flato et al., 2013).

We added a similar discussion to Section 4 in the manuscript.

<< About the minor comments >>

Q4: Table1: This table is very good and informative to provide the model's capability from the model-data comparison. It may be also helpful to add the deposition and burial fluxes, as described above in my comment (1).

A4: We added the burial fluxes to the table mentioned in A1b above.

Q5: Page 3 L31--33: This description is unclear. Does it mean that all burial fluxes return to the bottom water as dissolved properties? Please rewrite the description.

A5: No, the burial fluxes do not return to the bottom water as dissolved properties. The description is about the stack layers below the top reactive layer storing old deposits that are not reactive any longer in the model. The thickness of the reactive layer is always kept at 10 cm, and in case the net budget of solid

material reduces the thickness to below 10 cm, some old material from the stack layers will be "revived" to compensate for the loss in the reactive layer and to keep the 10-cm thickness.

We rephrased the relevant description to clarify that as follows: "The old deposits in the stack layers are treated as being not reactive any longer in the model. The thickness of the surface reactive layer is always kept at 10 cm, and in case the net budget of solid material reduces the thickness to less than 10 cm, some old material from the stack layers will be "revived" to compensate for the loss in the reactive layer and to keep the 10-cm thickness. "

Q6: Page 8 L5--6: This sentence is also unclear. Do you want to say that the ocean-sediment coupling is important to simulate the water properties? Please rephrase it to present your argument more clearly.

A6: We rephrased the sentence as follows: "Such large model errors would complicate the model–data comparison for the upper sediment composition. Therefore, the coupling of a more reliable sediment model like MEDUSA to CESM is essential for a straightforward comparison between model results and observations. "
(We assumed the reviewer had referred to P.8, L4--5)

Q7: Page 9 Line29: "over large areas" maybe mislead readers. In this paper, the sediment feedback is apparent in some regions, such as along the east coast of the equatorial Pacific, along the west coast of the Pacific, in the Arctic and Hudson Bay. Rather, the large difference in d13C in the North Atlantic arises from the model's bias in relation to AMOC or ocean mixing variability, which should be excluded from the sediment contributions to the bottom-water properties.

A7: We deleted the phrase "over large areas" from the sentence and modified it as follows. "In this study, the MEDUSA coupling produces $\delta^{13}C_{DIC}$ differences of up to 0.2‰ compared to the original BEC method through direct influence from the sediment and through feedbacks from the ocean physics leading to the water mass displacement as well."

Q8: Page 10 L29: provides -> provide

A8: It was corrected.

Reply to the Referee #2

We would like to thank the reviewer for the thoughtful and constructive comments.

Q1: Page 1 Line 16: I consider this sentence as a bit unclear or vague. Please state more explicitly which models might show a bias, and which results of those models might therefore be less reliable than previously thought.

A1: First, we replaced the word 'biases' with 'uncertainty' since we intended to address the general shortcomings of such models. Second, we added the following sentence to the abstract. "For example, an ocean model that does not treat sedimentary processes depending on the chemical composition of the ambient water can overestimate the amount of remineralization of organic matter in the upper sediment in an anoxic environment, which would lead to lighter $\delta^{13}C$ values in the bottom water."

Q2: Page 2 Line 24-25: "Up to the present, no fully-coupled comprehensive climate model has been coupled with a sediment diagenesis model for longer time-scale applications (e.g., the glacial-interglacial variations)." -> Has this approach been used on shorter (e.g. centennial) time scales? Can you give examples, and how does your approach differ from them?

A2: We revised the sentence as follows: "To our knowledge, a fully-coupled comprehensive climate model including a sediment diagenesis model has been applied to millennial time scales only (Jungclaus et al., 2010). We are planning to apply such a model to time scales of tens of millennia with the goal of simulating glacial–interglacial variations."

Q3: Page 5 Line 24-31: Can you show a figure, possibly in the online supplement, that proves that your time step was sufficiently small and your integration period sufficiently long to show something like a convergence of the sediment-water fluxes in the end (for all but the 14C of course)?

A3: In terms of DIC flux back to the ocean, the difference between a run with the original time step and a run with only 1/10 of the original time step is smaller than 0.5% for most grid cells. We added a figure to show this in the supplementary material. Time steps shorter than 1 year do not make sense because the input from CESM is annually averaged.

It should be noted that the CESM-MEDUSA coupled simulation (EXCPL) is not a "steady-state" run but a "transient" run where the model state evolves. Therefore, the length of the MEDUSA runs is not determined by the convergence of a model state but by the coupling interval.

Q4: Page 7 Line 3-4: "which would lead to the overestimate of biological production" -> "which would lead to an overestimation of biological production"?

A4: The sentence was corrected.

Q5: Page 9 Line 15-17: "Otherwise, one would need to translate records obtained from sediments into corresponding variables of the ocean model, which would introduce an additional source of uncertainty to the model--data comparison." -> You have an opposite translation by the MEDUSA model: ocean model variables are translated to sediment records. Why is this less uncertain than the other way around?

A5: The "forward modeling" by MEDUSA is beneficial because it provides a process-based translation rather than an empirical translation that would be inevitable without such a sediment model. Therefore, the point is not the direction of translation but the way of translation. We revised the sentence as follows. "Otherwise, one would need to translate records obtained from sediments into in an empirical way to corresponding variables of the ocean model, which would introduce an additional source of uncertainty to the model–data comparison."

Q6: Table 1: Would it make sense to add a third column for the values in the EXORG run?

A6: We added the following two tables (Tables 2 and 3 in the revised manusrcipt) to compare EXCPL and EXORG in Section 3.2 to show that the two experiments are comparable in general in terms of globally-integrated quantities (except for the burial flux). In Section 3.1, we would rather focus on the comparison between EXCPL and observations as in the current version of the manuscript.

**Table r1**. Globally-integrated annual mean deposition flux of particulate matter to the sediment and their burial flux (in parentheses) at the end of EXCPL and EXORG.

|  | EXCPL | EXORG |
| --- | --- | --- |
| POC (GtC/y) | 0.57 (0.091) | 0.51 (0.12) |
| $CaCO_3$ (GtC/y) | 0.39 (0.082) | 0.38 (0.14) |
| Opal (Tmol/y) | 46 (0.72) | 45 (3.4) |

**Table r2**. Total inventories in the global ocean of DIC, ALK, and PO$_4$ in EXCPL and EXORG. Values averaged over the last CESM run (10 surface years) are shown.

|                | EXCPL | EXORG |
|----------------|:-----:|:-----:|
| DIC (GtC)      | $3.660 \times 10^4$ | $3.657 \times 10^4$ |
| ALK (Peq)      | $3.201 \times 10^3$ | $3.201 \times 10^3$ |
| PO$_4$ (Pmol)  | 2.948 | 2.923 |

Q7: Figure 1: Why are the state variables only listed for the MEDUSA model and not for the BEC model? Probably the list of processes might be too long, but at least the state variables would give an indication of the model complexity for those not familiar with BEC.

A7: We updated the figure as follows.

[Figure]

Q8: Figure 6: Having this figure separate from Fig. 5 and using changed color scales makes the comparison very hard. And the improved behaviour of the model using the coupling is the main point of your manuscript. If you think the subfigures become too small if you put all three weight fractions for EXORG, EXCPL and OBS into one figure, you might consider one figure for each weight fraction but then containing EXORG, EXCPL and OBS?

A8: We re-arranged the figures according to the reviewer's suggestion.

[revised manuscript text omitted]